# Effects of Social Interaction and Depression on Homeboundness in Community-Dwelling Older Adults Living Alone

**DOI:** 10.3390/ijerph19063608

**Published:** 2022-03-18

**Authors:** You-Ri Kim, Hye-Sun Jung

**Affiliations:** Department of Preventive Medicine, College of Medicine, The Catholic University of Korea, 222 Banpo-daero, Seocho, Seoul 06591, Korea; yooul.k111@gmail.com

**Keywords:** older adults, homebound, social contact

## Abstract

This study examines the levels of social interaction, depression, and homeboundness, and the effects of social interaction and depression on homeboundness in community-dwelling older adults living alone. Survey data were collected from 6444 older adults aged 65 and over, living alone, who registered for individualized home care services at 42 public health centers in Gyeonggi Province. A total of 5996 participants with complete questionnaire data were included in the analysis. The mean social interaction score was 2.90 out of 6, and the mean depression score was 6.21 out of 15. The mean homeboundness score was 0.42 out of 2. A hierarchical multiple regression analysis was performed with general characteristics, health factors, social interaction, and depression to identify their effects on homeboundness. In general characteristics and health factors, homeboundness is associated with decreasing social interaction (β = 0.17, *p* < 0.001) and increasing depression (β = 0.25, *p* < 0.001) in older adults living alone. Homeboundness was severe among participants aged 80 and over (β = 0.04, *p* = 0.015) and those with several chronic diseases (β = 0.04, *p* < 0.001), falling history (β = 0.14, *p* < 0.001), and lack of exercise (β = −0.20, *p* < 0.001). Thus, interventions that target social interaction, depression, and health functions are important for this demographic.

## 1. Introduction

According to UN projections for the global population of 201 countries and the 2019 Global and Korean Population Trends and Projections by Statistics Korea, the older adult population (≥65 year) in Korea will grow at the fastest rate worldwide and reach 46.5% in 2067 [1,2]. The total number of households in Korea is gradually on the decline, but the proportion of single-person households is anticipated to continue to rise to 37.3% in 2047, 8.8% up from 2017. Furthermore, single-person households of adults aged 60 or older are projected to account for the greatest percentage of the population at 56.8% in 2047 [2]. As of 2017 (the year in which the study data were collected), there were 330,000 older adults aged 65 and over living alone in the Gyeonggi Province, accounting for 25% of the country’s total older adult single-person households [2,3]. The older adults living alone are more likely to be exposed to the risk of social isolation and loneliness due to the loss of their connection with family [4]. In addition, living alone for older adults affects the deterioration of their mental health and lowers their happiness. According to the 2010 Single-person households by Statistics Korea, older adults living alone have a low practice rate of nutritional status and healthy lifestyle, and a low health check-up rate for early detection of chronic diseases [5]. Therefore, it can be seen that living alone for older adults can have a negative effect on social isolation, psychological health, and physical health.

According to the 2017 National Survey of Senior Citizens by the Ministry of Health and Welfare (MOHW), the prevalence of physical chronic diseases and depressive symptoms is more than 10% higher in older adults aged 65 and over living alone compared to those in other types of households.

As shown here, older adults living alone are exposed to serious physical and mental health problems, and these factors are also determinants of “homeboundness”, the tendency to confine themselves at home [6].

The dictionary defines homeboundness as being confined at home without engaging in activities outside and refers to it as a state of staying at home in bed or a chair all day owing to mobility problems that hinder going out alone [7]. The definitions of “homeboundness” vary widely across studies. In addition to remaining in bed or a chair for most of or the entire day in the past two weeks [8], going out once or fewer times per week [9], and going out once or fewer times in the past month [10,11] are also included. In this study, homeboundness was defined as going out once or fewer times a week or a decreased frequency of going out compared to the preceding year [12].

Homeboundness lowers the likelihood of seeking healthcare, and thus, elevates the risk of serious health problems that require intensive care among older adults [13]. Thus, homeboundness increases mortality risk in older adults [14]. Moreover, it may increase the risk of solitary death, an emerging societal issue.

Homebound older adults are physically, mentally, and socially more vulnerable than their non-homebound counterparts. Previous studies have reported that homebound older adults are less mobile, have more chronic diseases, have a higher prevalence of cognitive impairment and severe depression, and have low social support compared to their non-homebound counterparts [15]. In particular, the homeboundness of older adults living alone requires dedicated attention. Studies have stated that older adults with little interaction with family or friends or older adults who live alone are more likely to be homebound [16], and that homeboundness elevates the risk of chronic diseases and disabilities in older adults living alone [17]. In particular, older adults living alone lack family connections and social support and have higher psychological anxiety and depression than others not living alone [8]. Older adults living alone experience disconnection from family and social networks, and are at high risk of economic poverty and isolation from support systems such as social insurance and public assistance [18]. In addition, older adults living alone are highly vulnerable due to poor nutrition and unhealthy lifestyle [5]. The social isolation, depression, and health-related vulnerabilities of the elderly living alone are on the continuum of the causal relationship that aggravates their homeboundness [19].

One key limitation of previous studies examining homeboundness was that they simply compared the rate of physical health problems, such as mortality and disability, between homebound and non-homebound groups, and these findings could not establish a causality between homeboundness and the factors that enable it, namely physical, mental, and social problems.

However, recent studies have shown that the prevalence of depression is higher in older adults, and in the homebound older adults living with depression, the factors affecting their depression are complicated, such as mobility restrictions or social isolation, unlike the non-homebound elders. Thus, that antidepressant treatment alone has a therapeutic effect is said to be limited [20]. In addition, it has shown that the decrease in physical function ability and social isolation affect the homeboundness of older adults [21].

Therefore, this study intends to identify the influencing factors that affect homeboundness, and to hierarchically identify the independent effects of demographic and health-related factors, social interaction, and depression on homeboundness.

To address these limitations of previous studies, this study investigated the independent effects of depression and social interaction on homeboundness in a hierarchical structure after surveying and controlling for the general characteristics of older adults living alone. According to this, it is intended to suggest the necessity of developing health and social management services to reduce homeboundness, which causes a health crisis for the elderly living alone.

## 2. Materials and Methods

### 2.1. Study Design

This study examines the levels of social interaction, depression, and homeboundness, along with the effects of social interaction and depression, on homeboundness in community-dwelling older adults living alone (Figure 1).

### 2.2. Study Population

Geographical characteristics of the population of Gyeonggi-do, the research target area, have the largest population in Korea. Therefore, it is a self-governing body representing the whole country [22]. However, the difference between the southern and northern regions of Gyeonggi-do is about three times larger and the regional variation is also large [22].

Of the 338,205 older adults (≥65 year) living alone in the Gyeonggi Province as of 2017, survey data from 6444 who consented to receive and were registered for individualized home care service provided by 42 public health centers in the Gyeonggi Province were considered. From these data, 5996 older adults (≥65 year) living alone with complete data for general characteristics, social interaction, depression, and homeboundness were included in the final analysis.

### 2.3. Study Duration and Ethical Considerations

The study data were collected by home-visiting nurses through a survey at the homes of participants in the Gyeonggi-do individualized home care service program from 1 May 2017 to 31 May 2017 after obtaining consent to participate in the survey for the elderly living alone registered at 42 public health centers in Gyeonggi-do. It was explained that if the subject wanted to cancel the service, they could withdraw the service registration and the questionnaire at any time.

This study was conducted by receiving data with personal information deleted from the encrypted file of the survey above from the research team in charge of the Gyeonggi Visiting Health Management Project. Encrypted data were stored on the researcher’s personal laptop, and security was maintained by setting a password so that the information would not be leaked to others.

This study was approved by the Institutional Review Board at Catholic University of Korea (MC21EASE0053).

### 2.4. Survey Content

#### 2.4.1. General Characteristics

Eight general characteristics were surveyed: age, length of solitary living, marital status, drinking, smoking, exercise (number of days that included walking for 10 min or more per week), fall history, and number of chronic diseases. The items were taken from the questionnaire for individualized home care service programs at public health centers.

#### 2.4.2. Social Interaction

The Social Interaction Scale for frail older adults for intensive care management developed by the MOHW was used [23]. This scale contains three items about interaction with family, friends, and opportunities to go out, and each item is rated on a 3-point Likert scale (0–2). The total score ranges from 0–6, with a score of 0–2 indicating extremely bad interaction, a score of 3–4 indicating bad interaction, and a score of 5–6 indicating good interaction [24]. The Cronbach’s alpha was 0.65 in this study.

#### 2.4.3. Depression (GDSSF-K)

The 15-item Geriatric Depression Scale Short Form-Korea Version (GDSSF-K) [25], a Korean-adapted and shortened version of the Geriatric Depression Scale (GDS) [26] developed by Yesavage et al., [26] was used. The Cronbach’s alpha of the scale was 0.88. A score of five or lower out of 15 was used to indicate a normal score, whereas a score of 6–9 and ten or higher was used to indicate mild depressive symptoms and severe depression, respectively.

#### 2.4.4. Homeboundness

To measure homeboundness, two items (items 16 and 17) out of the 25 items of the Kihon Checklist (KCL) developed by the Japanese Ministry of Health, Labor, and Welfare [27,28] were adapted into Korean by Seon et al., to “Do you go out at least once a week?” and “Do you go out less frequently compared to last year?” [12,19]. These items were also used in a 2007 survey of individual home care service programs for older adults aged 65 years and older, by the Korean Ministry of Health and Welfare (MOHW) [12,23]. The total score ranges from 0–2, with a score of 0 indicating not homebound and a score of 1 indicating mild homebound, a score of 2 indicating severe homebound. A higher score indicates more serious homeboundness [12]. The Cronbach’s alpha was 0.42 in this study.

### 2.5. Data Collection

The home care nurses of the individualized home care service program at 42 public health centers in Gyeonggi Province visited the participants in person between 1 May and 31 May 2017 to obtain written consent and administer the questionnaire 1:1. The survey was conducted only for those who agreed to the survey and service provision.

To enhance the reliability of the study data, a total of 115 surveyors, consisting of personnel from the Gyeonggi Provincial Government, a research team, and individualized home care service team members in 42 public health centers, were trained to guide the intervention questionnaire for the home care service program and data coding on 26 April 2017.

Data from 5996 out of 6444 older adults (≥65 year) were statistically analyzed using SPSS Version 25.0. The participants’ general characteristics were presented through frequencies, percentages, means, and standard deviations. The differences in homeboundness according to general characteristics, social interaction, and depression were analyzed with a *t*-test and an ANOVA, followed by the Scheffé post hoc test.

The effects of general characteristics, social interaction, and depression on homeboundness were analyzed using hierarchical multiple regression.

## 3. Results

### 3.1. Participants’ General Characteristics and Health Factors

The mean age was 80.4 (±5.87) year, and most of the participants (56.5%) were 80 year or older. The mean length of solitary living was 18.4 (±28.42) year, with the most common length of solitary living being 6–15 year (33.8%). The most common marital status was widowed (83.6%). The majority (85.8%) of the participants were non-smokers, and 72.7% reported not drinking alcohol. The most common “number of days of walking for 10 min or longer in a week” was 4–6 days (29.3%). Most participants (72.4%) had never had a fall, and the mean number of chronic diseases was 2.3 (±1.25). The mean depression score was 6.2 (±3.14) out of 15, and most participants (54.3%) had “mild depressive symptoms.” The mean homeboundness score was 0.4 (±0.65) out of 2, with most of the participants having a score of 0 (not homebound; 66.7%), and 24.6% having a score of 1 (mild homeboundness) (Table 1).

### 3.2. Degree of Homeboundness According to General Characteristics, Health Factors, Social Interaction, and Depression

Homeboundness was the highest among participants aged 80 or older (*p* < 0.001) and among those who lived alone for 30 year or longer (*p* < 0.01). In terms of exercise, homeboundness was the most serious among those who did not walk for 10 min or longer on any day of the week (*p* < 0.001), those with a fall history (*p* < 0.001), with 7–9 chronic diseases (*p* < 0.001), with extremely low social interaction scores (*p* < 0.001), and those with severe depression (*p* < 0.001) (Table 2).

### 3.3. Predictors of Homeboundness

Hierarchical multiple regression was performed to examine the effects of the participants’ general characteristics, health factors, social interaction, and depression on homeboundness. Prior to the analysis, multicollinearity among the study parameters was tested. Tolerance was above 0.1 and the variance inflation factor was below 10, confirming the absence of multicollinearity. The independence of the error term was tested using the Durbin–Watson test, with the results being within 0–4 at 1.828, confirming the absence of autocorrelation among the residuals.

Model 1 for examining the effects of general characteristics and health factors on homeboundness was statistically significant at F = 62.625 (*p* < 0.001). Age, exercise frequency, fall history, and number of chronic diseases were identified as the predictors of homeboundness, and these factors explained (*R*^2^) 11.2% of the variance. Homeboundness was severe in the ≥80 year group (β = 0.04, *p* = 0.015) and the lowest for those with 1–3 d of exercise (β = −0.01, *p* < 0.001), but severe among those with a fall history (β = 0.20, *p* < 0.001) and 7–9 chronic diseases (β = 0.06, *p* < 0.001).

Model 2 looked at the effect of social interaction on homeboundness after controlling for general characteristics and health factors (Model 1) and found it to be statistically significant at F = 77.123 (*p* < 0.001), and the F-change was statistically significant at 145.908 (*p* < 0.001). The percentage of explained variance (*R*^2^) was 15.3%, up by 4.1% from Model 1. The statistically significant predictors in Model 2 were identical to those in Model 1, and the explained variance was also similar. Homeboundness was severe when social interaction was “extremely bad” (β = 0.25, *p* < 0.001).

Model 3 examined the effects of depression on homeboundness after controlling for general characteristics, health factors, and social interaction. It was statistically significant at F = 105.304 (*p* < 0.001), and the F-change was also statistically significant at 423.556 (*p* < 0.001). *R*^2^ was 20.9%, up from 5.6% of Model 2. Model 3 showed that homeboundness was statistically significantly severe in those 80 year or older (β = 0.06, *p* < 0.001), and statistically significantly lower for those with 4–6 d of exercise (β = −0.20, *p* < 0.001) and fall history (β = 0.14, *p* < 0.001), and with 7–9 chronic diseases (β = 0.04, *p* < 0.001). Furthermore, severity of homeboundness was associated with “extremely bad” social interaction (β = 0.17, *p* < 0.001) and severe depression (β = 0.25, *p* < 0.001) (Table 3).

## 4. Discussion

The participants’ mean social interaction score was 2.90 out of 6, which is low, below the mean of 3. A past study on older women living alone reported low social interaction scores of 3–4, supporting this study’s findings [24]. Therefore, this study not only shows that older adults living alone have a lower than average social interaction score but also highlights the need to enhance their social interactions by implementing interventions that promote the domains of social interaction, such as interaction with family and opportunities for social involvement [24].

In this study, the mean depression score was 6.21 out of 15. This is higher than the score for a normal state (0–5) and indicates “mild depressive symptoms.” A previous study on older adults’ mental health according to solitary living reported that older adults living alone are likely to be socially isolated, and thus, display a high level of depression, supporting the current findings [29]. Furthermore, the onset of multiple chronic diseases and the consequent physical functional limitations may be socially isolating, and these factors may be responsible for the higher vulnerability to depression in this population [30]. In particular, older adults living alone have an inadequate social support system compared to their counterparts in other types of households, and for this reason, they are less likely to receive prompt treatment for physical or mental illnesses [13]. Thus, depression is highly likely to progress to severe mental disorders in older adults living alone compared to those who live with their families [15]. As older adults living alone are at greater odds of experiencing social isolation, loss, and depression, it is important to implement measures to manage their mental health.

In Model 1, an age of 80 or older, solitary living for 30 year or longer, 7–9 chronic diseases, fall history, and no days of walking for 10 min or longer in a week were identified as the predictors of homeboundness. This is consistent with previous findings in which the incidence of homeboundness is higher with age, poorer health and function, poorer instrumental activities of daily living [31], and in which the incidence of falls was higher among homebound older adults [32]. In addition, an Israeli longitudinal study on aging (CALAS) reported that factors that hinder walking, such as an increased number of stairs or the lack of an elevator, increase homeboundness, supporting the findings of this study [31]. Taken together, older age, having several chronic diseases, physical functional limitation, and increased fall risk or incidence curtail older adults’ opportunities to go out and may exacerbate their homeboundness [32].

Moreover, a lack of exercise or walking may lead to muscle weakening, limited ambulation, and physical frailty, which, in turn, discourage older adults from going out and elevates their risk of homeboundness [15]. As shown here, the results of this study show that such general characteristics elevate the risk for homeboundness. Therefore, we suggest that health care experts provide continuous interventions for exercise rehabilitation, such as muscle strengthening and stretching, for the elderly living alone because these programs can help solve the cause of homeboundness in some older adults living alone who have physical ailments and have limited activity due to falls.

The *R*^2^ of Model 2 was increased by 4.1% from Model 1, and this suggests that social interaction is a more potent predictor of homeboundness than general characteristics. It can be speculated that reduced social interaction due to curtailed contact with neighbors and the frequency of having visitors at home increases older adults’ homeboundness [33].

Older adulthood is a period in which individuals experience various losses, such as the death of their spouse or retirement, and for this reason, they are likely to have fewer social interactions and experience social isolation. In particular, older adults living alone have an inadequate support system that could buffer the impact of such losses. Social isolation resulting from a lack of a sense of belonging in a family or less frequent interactions with neighbors further thwarts their activity and emotionally intimidates older adults living alone, thus increasing their homeboundness [34]. Increased homeboundness among older adults living alone has an adverse impact on their health. This can be explained by the possibility that homebound older adults are less likely to seek regular health care before developing chronic diseases, and this not only induces serious health problems requiring intensive care [13] but also delays treatment, thereby increasing their mortality risk [35]. Thus, it is important to implement measures to promote social activities, such as interaction with families or neighbors, and religious activities, for older adults living alone to bolster their social interactions and reduce homeboundness. For this reason, health managers should visit older adults living alone and understand they are vulnerable to missing out on community networks and information [36]. In addition, it is necessary to provide information and education on social support systems and services in an easy to understand manner. For example, it is necessary to guide them toward local gathering places that are convenient to access even for the elderly, such as neighborhood parks, churches, senior centers, and community centers, and to play a role in facilitating intimacy and social exchange through meeting activities [36].

The *R*^2^ of Model 3 was 5.6% higher than that of Model 2. This suggests that depression is a more potent predictor of homeboundness than general characteristics or social interaction in older adults living alone. A previous study that compared the prevalence of psychiatric diseases between homebound and non-homebound older adults after controlling for their health status reported that the risk of psychiatric disorders, such as depression, mood disorders, and anxiety disorders, among homebound older adults is twofold that of non-homebound older adults [8], and that depression is an independent predictor of homeboundness in this demographic, supporting the findings of this study [37]. Diminished motivation or reduced activity are some of the psychological factors that influence homeboundness in older adults. A depressed mood exacerbates homeboundness by reducing motivation and activity [38]. Thus, it is necessary to reduce homeboundness through health interventions that alleviate depressive symptoms in older adults living alone. In essence, managing social interaction and depression is expected to reduce homeboundness in older adults living alone. Specifically, it is important that nurses visit to monitor whether the depressed older adults living alone are receiving antidepressant and chronic disease medications and treatment. Moreover, if necessary, connecting with organizations that provide professional mental health services in the community, which are available free of charge, such as local mental health promotion centers or psychological support centers, will help alleviate depression and reduce homeboundness. The reason is that in such an institution, forming a group and creating a community with geriatric depression, can help build social interaction and emotional consensus. Therefore, these alternatives may help older adults living alone suffering from depression to be relieved of their feelings of alienation and isolation and to go out of the house.

The limitations of this study pertains to its cross-sectional design, which hinders establishing temporal precedence between the cause and effect. Hence, longitudinal studies are needed to establish a temporal causality among social interaction, depression, and homeboundness. In addition, the study data were collected from 42 public health centers, so the findings of this study cannot be generalized to the entire population of older adults living alone.

Despite these limitations, however, one strength of this study is that it sheds light on the effects of social interaction and depression on homeboundness among older adults living alone, who may be physically and mentally more vulnerable than other older adults. Furthermore, the surveyors were trained and given survey guidelines prior to administering the surveys, and experienced home care nurses administered the surveys themselves in person, adding to the reliability of the data. Another strength of this study is that, in contrast to previous studies that classified the participants into homebound and non-homebound groups and compared the prevalence of physical and mental illnesses between the two groups, this study attempted to quantify homeboundness and examined general characteristics and psychosocial factors that have been previously associated with the level of homeboundness in community-dwelling older adults living alone, to shed light on the priorities among the major predictors of homeboundness.

Thus, the results of this study can serve as evidence for community health projects that seek to improve social interactions and manage depression in community-dwelling older adults living alone and ultimately contribute to alleviating homeboundness in this demographic.

## 5. Conclusions

The results of this study imply at least three general conclusions. First, social interaction was a more potent predictor of homeboundness than general characteristics in older adults living alone, so health management interventions for this group of older adults should incorporate activities that promote social interaction in order to decrease homeboundness.

Second, depression was a more potent predictor of homeboundness than general characteristics and social interaction in older adults living alone, suggesting that it is a stronger factor. Therefore, depression should be set as the priority target of interventions in order to prevent homeboundness in older adults living alone, and the intervention programs should also include activities that promote social interaction. Third, homeboundness was higher with older age (≥80 year) and with multiple chronic diseases among older adults living alone, so it is important to periodically monitor and intervene in their health functional status, social interaction, depression, and level of homeboundness to properly address the issue.

## Figures and Tables

**Figure 1 ijerph-19-03608-f001:**
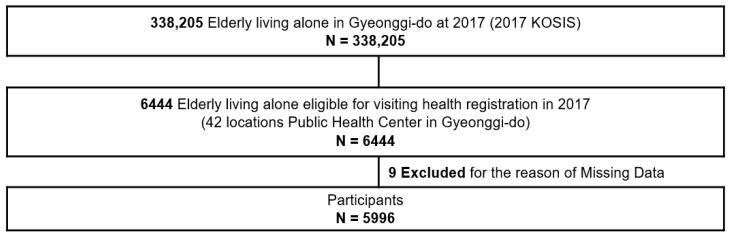
Study population.

**Table 1 ijerph-19-03608-t001:** General characteristics and health factors, social interaction, depression, homeboundness.

			*N* = 5996
Variable	Categories	*N* (%)	M ± SD
Age			80.4 ± 5.87
	65–74	941 (15.7)	
	75–79	1665 (27.8)	
	80 above	3390 (56.5)	
Period of Living Alone			18.4 ± 28.42
	0–5 less than a year	1184 (19.7)	
	6–15	2026 (33.8)	
	16–29	1406 (23.4)	
	30 more than a year	1380 (23.1)	
Marital status			
	Single	203 (3.4)	
	Divorce	578 (9.6)	
	Widowed spouse	5014 (83.6)	
	Separation	136 (2.3)	
	Others	65 (1.1)	
Smoking			
	Yes	871 (14.5)	
	No	5125 (85.8)	
Drinking			
	Yes	1639 (27.3)	
	No	4357 (72.7)	
Number of days of Exercise (walking for 10 min or longer in a week)			
	*n*	1306 (21.8)	
	1–3	1467 (24.5)	
	4–6	1756 (29.3)	
	daily	1467 (24.5)	
Experience of Falling			
	Yes	1656 (27.6)	
	No	4340 (72.4)	
Number of Chronic diseases			2.3 ± 1.25
	<1	225 (3.8)	
	1–3	4815 (80.3)	
	4–6	924 (15.4)	
	7–9	32 (0.5)	
Social Interaction			2.9 ± 1.61
	Good	1019 (17.0)	
	Bad	2748 (45.8)	
	Extremely bad	2229 (37.2)	
Degree of Depression			6.2 ± 3.14
	Severe	929 (15.5)	
	Mild	3257 (54.3)	
	Normal	1810 (30.2)	
Degree of Homeboundness			0.4 ± 0.65
	Severe	523 (8.7)	
	Mild	1473 (24.6)	
	None	4000 (66.7)	

**Table 2 ijerph-19-03608-t002:** Homeboundness status according to general characteristics, health factors, social interaction, and depression of elderly living alone.

					*N* = 5996
Variable	Categories	*N* (%)	Homeboundness		
			M ± SD	t/f (*p*)	Scheffé
Age				14.07 (<0.001)	a, b < c
	65–74 ^a^	941 (15.7)	0.38 ± 0.62		
	75–79 ^b^	1665 (27.8)	0.36 ± 0.61		
	≥80 ^c^	3390 (56.5)	0.46 ± 0.67		
Period of Living Alone (year)				4.37 (0.004)	a < c, d
	0–5 ^a^	1184 (19.7)	0.36 ± 0.61		
	6–15 ^b^	2026 (33.8)	0.42 ± 0.65		
	16–29 ^c^	1406 (23.4)	0.44 ± 0.66		
	≥30 ^d^	1380 (23.0)	0.44 ± 0.65		
Marital Status				1.80 (0.126)	
	Single	203 (3.4)	0.51 ± 0.65		
	Divorce	578 (9.6)	0.46 ± 0.647		
	Widowed spouse	5014 (83.6)	0.41 ± 0.65		
	Separation	136 (2.3)	0.40 ± 0.65		
	Others	65 (1.1)	0.35 ± 0.60		
Smoking				1.82 (0.059)	
	Yes	871 (14.5)	0.45 ± 0.65		
	No	5125 (85.5)	0.41 ± 0.64		
Drinking				−0.52 (0.264)	
	Yes	1639 (27.3)	0.41 ± 0.64		
	No	4357 (72.7)	0.42 ± 0.65		
Number of days of Exercise (walking for 10 min or longer in a week)				121.60 (<0.001)	a > b > c, d
	No ^a^	1306 (21.8)	0.67 ± 0.79		
	1–3 ^b^	1467 (24.5)	0.49 ± 0.66		
	4–6 ^c^	1756 (29.3)	0.28 ± 0.52		
	Daily ^d^	1467 (24.5)	0.30 ± 0.54		
Experience of Falling				15.52 (<0.001)	
	Yes	1656 (27.6)	0.64 ± 0.72		
	No	4340 (72.4)	0.33 ± 0.59		
Number of Chronic diseases				26.28 (<0.001)	a, b, c < d
	<1 ^a^	225 (3.8)	0.44 ± 0.66		
	1–3 ^b^	4815 (80.3)	0.39 ± 0.63		
	4–6 ^c^	924 (15.4)	0.55 ± 0.70		
	7–9 ^d^	32 (0.5)	1.06 ± 0.95		
Social Interaction				222.32 (<0.001)	a < b < c
	Good	1019 (17.0)	0.21 ± 0.45		
	Bad	2748 (45.8)	0.32 ± 0.55		
	Extremely bad	2229 (37.2)	0.64 ± 0.77		
Degree of Depression				317.42 (<0.001)	a > b > c
	Severe ^a^	929 (15.5)	0.83 ± 0.80		
	Mild ^b^	3257 (54.3)	0.42 ± 0.80		
	Normal ^c^	1810 (30.2)	0.21 ± 0.46		

a–d: scheffe’post-hoc test (same letter not significient).

**Table 3 ijerph-19-03608-t003:** The effect of general characteristics, health factors, social interaction, and depression on the homeboundness status of elderly living alone.

*N* = 5996											
Variable	Model 1			Model 2			Model 3				
	Homeboundness			Homeboundness			Homeboundness				
	*β*	*t*	*p*	*β*	*t*	*p*	*β*	*t*	*p*	*TOL*	VIF
Constant		17.26	<0.001		19.55	<0.001		9.72	<0.001		
Age											
(Ref: 65–74)											
75–79	−0.01	−0.50	0.620	0.00	0.15	0.878	0.01	0.82	0.414	0.49	2.01
>80	0.04	2.43	0.015	0.05	2.71	0.007	0.06	3.62	<0.001	0.49	2.02
Period of living alone (year)											
(Ref: 0–5)											
6–15	0.02	1.26	0.208	0.01	0.90	0.366	0.01	0.83	0.407	0.55	1.81
16–29	0.03	1.66	0.097	0.02	1.16	0.247	0.02	0.98	0.328	0.59	1.69
>30	0.03	2.02	0.043	0.02	0.98	0.329	0.01	0.79	0.429	0.58	1.70
Number of days of Exercise (walking for 10 min or longer in a week)											
(Ref: No)											
1–3	−0.01	−8.74	<0.001	−0.11	−7.52	<0.001	−0.10	−6.47	<0.001	0.61	1.63
4–6	−0.27	−17.04	<0.001	−0.23	−14.61	<0.001	−0.20	−13.03	<0.001	0.58	1.72
Daily	−0.20	−15.11	<0.001	−0.19	−12.61	<0.001	−0.17	−11.07	<0.001	0.59	1.67
Experience of Falling											
(Ref: No)											
Yes	0.20	15.99	<0.001	0.18	15.21	<0.001	0.14	11.91	<0.001	0.95	1.06
Number of Chronic diseases											
(Ref: <1)											
1–3	−0.06	−2.28	0.023	−0.05	−1.81	0.070	−0.04	−1.44	0.150	0.22	4.44
4–6	0.01	0.52	0.603	0.02	0.87	0.385	0.01	0.47	0.638	0.22	4.37
7–9	0.06	4.32	<0.001	0.05	4.09	<0.001	0.04	3.27	0.001	0.87	1.15
Social Interaction											
(Ref: Good)											
Extremely bad				0.25	14.58	<0.001	0.17	9.86	<0.001	0.45	2.20
Bad				0.07	3.98	<0.001	0.03	1.64	0.102	0.49	2.05
Depression							0.25	20.58	<0.001	0.87	1.15
*R* ^2^	0.112			0.153			0.209				
Adj. *R*^2^	0.110			0.151			0.207				
Change *R*^2^				0.041			0.056				
F (*p*)	62.625 (<0.001)			77.123 (<0.001)			105.304 (<0.001)				
Change F (*p*)				145.908 (0.001)			423.556 (<0.001)				
Durbin–Watson	1.828										

## Data Availability

Not applicable.

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
