# Peer review of "Effects of Social Interaction and Depression on Homeboundness in Community-Dwelling Older Adults Living Alone"

_ijerph, 2022, doi:10.3390/ijerph19063608_

Round 1

Reviewer 1 Report

Thank you for this opportunity for reviewing this manuscript. This is a study covering multiple research areas, including mental health, social epidemiology, and demographics thus the findings of this study can contribute to other researchers, clinicians, and policy makers in those fields. As the authors mentioned in the manuscript, the cross-sectional design can be a limitation. However, it is not generally feasible to produce definitive conclusions on causality in this area of research due to the complexity of the constructs of mental health. Therefore, I perceived the current study presents meaningful findings that can be foundations for future research in other aging countries could be rather a strength of the present study.

I would recommend authors to discuss how authors could state “effect” of social interaction and depression on homebound not “association” or “relation” referring previous studies.

Author Response

Thank you very much for reviewing my thesis and for giving me a good evaluation. We will respond to your comments as follows.

Point 1: I would recommend authors to discuss how authors could state “effect” of social interaction and depression on homebound not “association” or “relation” referring previous studies.

Response 1: A previous study on the effects of depression and social isolation on homeboundness was added. In addition, in the case of the elderly suffering from depression, a study was attached that indicated that not only antidepressant treatment but also homeboundness was required.

Reviewer 2 Report

ijerph-1589328

General comment

The manuscript entitled “Effects of Social Interaction and Depression on Homeboundness in Community-dwelling Older Adults Living Alone”, which aimed to assess the factors associated with homeboundness in a relatively large population of older adults living alone. Please find my comment below.

Abstract

  1. Lines 16-18 “General characteristics explained 11.2% of the variance. Controlling for general characteristics, social interaction explained 15.3% of the variance, and controlling for both general characteristics and social interaction, depression explained 20.9% of the variance.” – I would suggest authors remove these statement from abstract and include the statistic results (beta, CI and p-value) for each variables (social isolation and depression) in the next sentence to guide the conclusion of the study.

Background

  1. lines 44-59 – Please avoid writing one sentence per one paragraph
  2. Lines 69-73 – one key limitation of previous studies ….. these findings could not establish a a causality between homeboundness and the factors that enable it, namely physical, mental, and social problems. – from this statement, I assume that the physical health issues were also assessed in the study. The objective in the next paragraph should state it clearly that the physical health status was included (not only general characteristics).
  3. Lines 74-77 – As the main purpose of the study was to assess the independent effects of social interaction and depression on homeboundness, please add more detail about the current knowledge about social isolation and depression in association with homeboundness.

Methods

  1. Lines 87-93 – under the study population subsection, please add the detail about how the older adults enter to this individualized home care service. Were they recruited from the community by survey, or they can apply by themselves? What are the criteria? So readers can have more insight about study setting.
  2. Lines 102-106 – The data about general characteristics, was it obtained from the electronic medical record or from the face-to-face interview in May 2017. If it was from the medical record, I guess there were more than one visit per person so the data in which visit was used in the analysis. Please clarify.
  3. Lines 109-114 – Social interaction scale was categorized into 3 categories. (extremely low, low, and good). However, in Tables, they were extremely bad, bad, and good. Please use the same words throughout the manuscript to prevent confusion.
  4. Lines 125-133 – please also add the detail about how to define the degree of homeboundness as high and low (these words were stated in Table 1).

Results

  1. Page 6 (Table 1) - Degree of depression --> Sever”e”
  2. Lines 162-163 “having a score of 1 (moderate homeboundness)” – again, please make sure that the authors use the same word throughout the manuscript. Moderate homeboundness has never been mentioned before and not in Table 1 too.
  3. Page 8 (Table2) - degree of depression --> high / low / none . Please consider changing those words to severe / mild/ normal as used earlier.

Discussion

  1. It would be nice to see the implementation of the results in real life practice. The author can provide sum examples of those interventions that were mentioned in discussion section including intervention to reduce depression or improve social isolation.
  2. Lines 282 – “One limitation” can be change to “Limitations” as the authors mentioned more than one limitation.

Author Response

Thank you for your detailed review of my manuscript. It has been modified as follows. 

Point 1: Lines 16-18 “General characteristics explained 11.2% of the variance. Controlling for general characteristics, social interaction explained 15.3% of the variance, and controlling for both general characteristics and social interaction, depression explained 20.9% of the variance.” – I would suggest authors remove these statement from abstract and include the statistic results (beta, CI and p-value) for each variables (social isolation and depression) in the next sentence to guide the conclusion of the study.

Response 1: Lines 16-18 were deleted and the statistical results were recorded in the abstract.

  • Homeboundness is associated with decreasing social interaction (β = 0.17, p<.001) and increasing depression (β = 0.25, p<.001) in older adults living alone. Homeboundness was severe among participants aged 80 (β = 0.04, p=.015) and over and those with several chronic diseases (β = 0.04, p<.001), falling history (β = 0.14, p<.001), lack of exercise (β = -0.20, p<.001). Thus, interventions that target social interaction, depression, and health functions are important for this demo-graphic.

Point 2: lines 44-59 – Please avoid writing one sentence per one paragraph

Response 2: I reflected.

Point 3: Lines 69-73 – one key limitation of previous studies ….. these findings could not establish a a causality between homeboundness and the factors that enable it, namely physical, mental, and social problems. – from this statement, I assume that the physical health issues were also assessed in the study. The objective in the next paragraph should state it clearly that the physical health status was included (not only general characteristics).

Response 3: I reflected.

  • However, recent studies have shown that prevalence of depression is higher in the older adults, For the homebound older adults suffering from depression, there s a limit to the therapeutic effect of only antidepressant treatment because mobility restrictions or so-cial isolation affect depression. In addition, the decrease in the ability to function and so-cial isolation affect the homeboundness of the older adults. Therefore, this study intends to identify the influencing factors that affect homeboundness. it is intended to hierarchically identify the independent effects of demographic and health-related factors, social interac-tion, and depression on homboundness.

Point 4: Lines 74-77 – As the main purpose of the study was to assess the independent effects of social interaction and depression on homeboundness, please add more detail about the current knowledge about social isolation and depression in association with homeboundness.

Response 4: I reflected.

Point 5: Lines 87-93 – under the study population subsection, please add the detail about how the older adults enter to this individualized home care service. Were they recruited from the community by survey, or they can apply by themselves? What are the criteria? So readers can have more insight about study setting.

Response 5:

Point 6: Lines 102-106 – The data about general characteristics, was it obtained from the electronic medical record or from the face-to-face interview in May 2017. If it was from the medical record, I guess there were more than one visit per person so the data in which visit was used in the analysis. Please clarify.

Response 6: I reflected

  • The study data were collected by home-visiting nurses through a survey at the homes of participants in the Gyeonggi-do individualized home care service program from May 1, 2017 to May 31, 2017. The survey was conducted only for those who agreed to the survey and service provision. It was explained that if the subject wanted to terminate the service, he would withdraw from the service registration at any time and withdraw the questionnaire.
  • This study was conducted by receiving data that deleted personal information from the coded file of the above survey from the research team in charge of the Gyeonggi-do Visiting Health Management Project. The coded data was stored in the researcher's personal laptop and security was maintained by setting a password so that the information would not be leaked to others.

Point 7: Lines 109-114 – Social interaction scale was categorized into 3 categories. (extremely low, low, and good). However, in Tables, they were extremely bad, bad, and good. Please use the same words throughout the manuscript to prevent confusion.

Response7: Unifed. (extremely bad interaction, a score of 3–4 indicating bad interaction, and a score of 5–6 indicating good interaction in manuscript and Tables.)

Point 8: Lines 125-133 – please also add the detail about how to define the degree of homeboundness as high and low (these words were stated in Table 1).

Response 8:  score 1: mild homeboundness, score 2: severe homeboundness

Point 9: Page 6 (Table 1) - Degree of depression --> Sever”e”

Response 9: I reflected

Point 10: Lines 162-163 “having a score of 1 (moderate homeboundness)” – again, please make sure that the authors use the same word throughout the manuscript. Moderate homeboundness has never been mentioned before and not in Table 1 too.

Response 10: I Edited -  ‘mild homeboundness’

Point 11: Page 8 (Table2) - degree of depression --> high / low / none . Please consider changing those words to severe / mild/ normal as used earlier.

Response 11: changed those words to severe / mild/ normal in Table2.

Point 12: Discussion

It would be nice to see the implementation of the results in real life practice. The author can provide sum examples of those interventions that were mentioned in discussion section including intervention to reduce depression or improve social isolation.

Response 12 : In the discussion, the Global age-friendly cities: a guide presented by the WHO was referred to, and cases were presented for symptom management and management that can enhance social interaction.

Point 13: Lines 282 – “One limitation” can be change to “Limitations” as the authors mentioned more than one limitation.

Response 13: I Edited

Reviewer 3 Report

Thank you for the opportunity to review this paper.  The following comments are hopefully designed to help you make alterations to the paper to improve its relevance to readers.  On the whole I feel this is a good paper which adds well to the evidence base, with some minor issues.

Paragraph lines 36-43: There is a great deal of literature charting the varied impacts of social isolation that I would like to see you use here, as I feel limiting it to just issues like suicidal ideation and solitary deaths is a little extreme.

Line 50: while the definition of homeboundness is useful, I think more depth on the potential causes of such issues from the literature would help (e.g. loss of family and social connections, increased frailty).  This is discussed later as a result of homeboundness, but it can also be a cause.

Line 69: There is also qualitative work on social isolation that gives some context that could be drawn from, as that experience is important alongside "harder" health outcomes.

Line 87: A description of the province is needed here, in terms of socio-cultural characteristics, to give context to the findings.

Line 150: There is no discussion of ethical issues here, which is interesting given the high response rate.  Can you discuss how the participants constented to have their data collected?

Line 244: While I accept the point that targeted interventions are required, I feel like this is a fairly obvious suggestion - are there recommendations of specific interventions which would seem to work?

Line 266: Again, I feel like drawing from wider literature would help here - concepts like age-friendly cities are designed to support interactions between generations. Considering them here would add depth to the findings.

Line 282: I think the focus on one setting should also be considered a limitation, particularly if you can describe the province in depth earlier.

Author Response

Thank you for your kind review and review of my article.

I have responded to your comment as follows.

Point 1:. Paragraph lines 36-43: There is a great deal of literature charting the varied impacts of social isolation that I would like to see you use here, as I feel limiting it to just issues like suicidal ideation and solitary deaths is a little extreme.

Response 1: I edited with reference.

  • The older adults living alone are more likely to be exposed to the risk of social isolation and loneliness due to loss of connection with family. In addition, living alone in the el-derly affects the deterioration of mental health and lowers their happiness. According to the 2017 Single-person households by statistics Korea, The older adults living alone have a low nutritional status and healthy lifestyle practice rate and a low health check-up rate for early detection of chronic diseases.

Point 2: Line 50: while the definition of homeboundness is useful, I think more depth on the potential causes of such issues from the literature would help (e.g. loss of family and social connections, increased frailty).  This is discussed later as a result of homeboundness, but it can also be a cause.

Response 2: I edited with reference.

  • In particular, older adults living alone lack family ties and social support and have higher psychological anxiety and depression than the elderly living alone . Older adults living alone experience reduced family and social networks and are at high risk of eco-nomic poverty and isolation from support systems such as social insurance and public assistance . In addition, older adults living alone are highly vulnerable due to poor nutri-tion and healthy lifestyle practices . The social isolation, depression, and health-related vulnerabilities of the elderly living alone are on the continuum of the causal relationship that aggravates their homeboundness .
  •  

Point 3: Line 69: There is also qualitative work on social isolation that gives some context that could be drawn from, as that experience is important alongside "harder" health outcomes.

Response 3: I referenced. (Kwon, H.C. A Qualitative Study on the Social Isolation and Poverty of the Elderly living alone. THE JOURNAL OF SOCIAL SCIENCE 26(3), 2019.9, 135-160(26 pages)

http://www.dbpia.co.kr/journal/articleDetail?nodeId=NODE09216498)

Point 4: Line 87: A description of the province is needed here, in terms of socio-cultural characteristics, to give context to the findings.

Response 4: I edited.

  • Geographical characteristics of the population of Gyeonggi-do, the research target area, have the largest population in Korea. Therefore, it is a self-governing body repre-senting the whole country. However, the difference between the southern and northern re-gions of Gyeonggi-do is about three times larger and the regional variation is also large.

Point 5: Line 150: There is no discussion of ethical issues here, which is interesting given the high response rate.  Can you discuss how the participants constented to have their data collected?

Response 5: I edited.

Point 6: Line 244: While I accept the point that targeted interventions are required, I feel like this is a fairly obvious suggestion - are there recommendations of specific interventions which would seem to work?

Response 6: I changed sentances.

  • Therefore, we suggest that health care experts provide continuous interventions for exercise rehabilitation, such as muscle strengthening and stretching, for the elderly living alone. Because these programs can help solve the cause of homeboundness in some older adults living alone who have physical ailments and have limited activity due to falls.
  •  

Point 7: Line 266: Again, I feel like drawing from wider literature would help here - concepts like age-friendly cities are designed to support interactions between generations. Considering them here would add depth to the findings.

Response 7: In the discussion, the Global age-friendly cities: a guide presented by the WHO was referred to, and cases were presented for symptom management and management that can enhance social interaction.

Point 8: Line 282: I think the focus on one setting should also be considered a limitation, particularly if you can describe the province in depth earlier.

Response 8: : I edited.
